# Progress in the Preparation and Modification of Zinc Ferrites Used for the Photocatalytic Degradation of Organic Pollutants

**DOI:** 10.3390/ijerph191710710

**Published:** 2022-08-28

**Authors:** Jinyuan Zhu, Yingying Zhu, Zhen Chen, Sijia Wu, Xiaojian Fang, Yan Yao

**Affiliations:** 1Faculty of Maritime and Transportation, Ningbo University, Ningbo 315211, China; 2College of Metrology & Measurement Engineering, China Jiliang University, Hangzhou 310018, China

**Keywords:** zinc ferrite, catalyst, photochemistry, nanomaterials, wastewater

## Abstract

Zinc ferrite is a type of photocatalytic material with high physicochemical stability, narrow band gap, high carrier separation efficiency, high porosity, and paramagnetism, which makes it easy to recover. Thus, zinc ferrite is widely used as a photocatalyst in water treatment. In this paper, the preparation principles as well as the advantages and disadvantages of typical methods used to prepare zinc ferrite including hydrothermal, co-precipitation, sol-gel, and other novel methods such as biosynthesis have been summarized. Modification methods such as elemental doping, composite formation, and morphological modification have been highlighted. Using these modification methods, the catalytic activity of zinc ferrite toward the photocatalytic degradation of organic pollutants in water has been enhanced. Biosynthesis is regarded as a promising preparation method that uses biological materials instead of chemical materials to achieve the large-scale preparation of zinc ferrite using low cost, energy efficient, and environmentally friendly processes. Meanwhile, the combination of multiple modification techniques to enhance the photocatalytic performance of zinc ferrite will be an important research trend in the future.

## 1. Introduction

The rapid development of industrialization and urbanization over the past few decades, which is due to the rapid growth of the global population, has led to an increase in energy consumption and a series of environmental issues. Wastewater discharged during industrial processes generally contains complex compositions including organic dyes (carmine, rhodamine B (RhB), methyl blue (MB)), phenols (phenol, bisphenol A), pesticides (dimethyl dichlorovinyl phosphate, dichloro diphenyl trichloroethane, 4-chlorophenol), antibiotics (tetracycline hydrochloride, terramycin), etc. Such chemical substances have strong water solubility, good photostability, and thermal stability, and are difficult to degrade naturally. Thus, the direct discharge of such wastewater will lead to changes in the water color, inhibition in plant growth, human cancer, etc., which are threatening aquatic ecosystems and human beings [1]. For example, basic dyes can irritate the skin and induce cancer [2], phenol can cause cardiovascular disease [3], and tetracycline can destroy the respiration and digestion of soil microorganisms [4]. In order to meet the environmental standards, wastewater must be further treated with pollutant degradation materials in order to remove the harmful pollutants present. Various wastewater treatment methods have been developed including adsorption [5,6], coagulation [7,8], sedimentation [9,10], membrane technologies [11,12], advanced oxidation process [13,14], and biological approaches [15,16].

As one of the most advanced oxidation processes, photocatalysis is currently at the forefront of laboratory research. The principle of photocatalysis is that the photocatalyst produces redox capacity under solar light irradiation to achieve the purpose of degrading pollutants [17,18], reducing CO_2_ [19], or splitting water to produce hydrogen [20,21]. Semiconductors, which are low in cost and high in photocatalytic efficiency, are usually used as catalysts in photocatalytic oxidation reactions. At present, the nanoscaled semiconductor photocatalyst, which has a unique size, surface, grain boundary, and quantum tunneling effects, is one of the most active research areas in photocatalysis. Nanomaterials have several advantages over traditional pollutant degradation materials including good solubility, large specific surface area, abundant active sites, short particle diffusion distances, and good adsorption properties [22]. The active sites in the catalyst can be fully utilized due to the excellent dispersion in pollutant solutions and the high adsorption capacity of nanomaterials. These properties provide an efficient optimization method for carrier separation and charge transfer behavior inside the catalyst [23]. However, there are problems associated with nanomaterials such as facile agglomeration and difficulty in recycling. The unique magnetic properties of spinel ferrite make it an extremely useful nanomaterial for various applications such as medical devices [24,25,26], electronic components [27], and sewage treatment [28,29].

Spinel ferrite has a face-centered cubic crystal structure, which belongs to the Fd-3m space group. Its general formula is MFe_2_O_4_, where M represents a divalent metal ion (Co, Zn, Mg, Ni, Ba, and Mn, etc.) [30]. Figure 1 shows that spinel ferrite is classified into normal spinel, inverse spinel, and complex spinel based on its crystal structure [31]. There are eight molecular units in the cubic cell of spinel ferrite, namely M_8_Fe_16_O_32_. Normally, there are 96 voids between the 32 oxygen anions in the cubic dense array of the spinel crystals. There are eight divalent cations in the 64 tetrahedral A sites, while Fe^3+^ occupies half of the 32 octahedral B sites. Inverse spinel structures have Fe^3+^ ions occupying the tetrahedral sites and other metal cations occupying the octahedral sites [32,33,34]. The complex spinel structure consists of M^2+^ and Fe^3+^ randomly occupying tetrahedral and octahedral sites with the structural formula (M^2+^_1−λ_Fe^3+^_λ_) (M^2+^_λ_Fe^3+^_2−λ_) O^2−^_4_, where λ indicates the degree of inversion [35,36]. When the structure changes, metal ions begin to occupy the inversion site and the magnetic behavior changes to ferromagnetism. This cation rearrangement results in the formation of two magnetic sublattices, which further leads to changes in the magnetization value of zinc ferrite (ZnFe_2_O_4_) [37].

ZnFe_2_O_4_ usually has a normal spinel structure with strong chemical bonds, tight atomic stacking, and good stability. In the past, the crystal structure of ZnFe_2_O_4_ at the nanoscale was thought to be a complex spinel structure (i.e., (Zn^2+^_1−λ_Fe^3+^_λ_) (Zn^2+^_λ_Fe^3+^_2−λ_) O^2−^_4_ [37,38]). In recent years, a crystal structure that is composed of a single-phase mixture of normal spinel and inverse spinel crystals has also been proposed [39,40].

As with all spinel ferrites, ZnFe_2_O_4_ shows high photocatalytic activity, and is easy to prepare and recover, while Zn^2+^ greatly improves its photosensitive properties, gas-sensitive properties, wave absorption properties, and antioxidant capacity [41,42,43]. ZnFe_2_O_4_ has a Curie temperature of 375 °C and a Néel temperature of −263 °C [44,45]. Consequently, ZnFe_2_O_4_ is stable and exhibits excellent paramagnetic properties in conventional preparation processes (e.g., hydrothermal synthesis, usually below 250 °C) and working conditions (room temperature, 20 °C). In addition, ZnFe_2_O_4_ has good photochemical stability, high porosity, and narrow band gap (1.9 eV), which makes it a good candidate as a photocatalyst [46,47]. ZnFe_2_O_4_ is now widely used in organic dye degradation [48], heavy metal recovery [49], photoelectric fuel cell [50], antibiotic degradation [51], etc., and has good reusability [52]. ZnFe_2_O_4_ can be extracted from industrial waste with low cost, which is conducive to its massive production [53].

ZnFe_2_O_4_ is a typical N-type iron-based semiconductor, which has free electron conduction bands with higher energy and a full electron valence band with a definite band gap. When the photon energy is greater than the semiconductor’s band gap, the electrons move from the valence band to the conductive band (CB) of the semiconductor, while the generated holes remain in the valence band (VB), forming electron–hole pairs (see Equation (1)). The excited electron can either recombine with a hole (see Equation (2)) or react with an adsorbed species (see Equation (3)) [54]. Hydroxyl radicals can be produced by photocatalytic oxidation (see Equation (4)). Organic pollutants react with hydroxyl radicals to generate intermediates (see Equation (5)), which in turn reacts with hydroxyl radicals to form mineral salts, carbon dioxide, and water (see Equation (6)), while superoxide radicals can be formed by the electrons in the conduction band reacting with oxygen-containing molecules (see Equation (7)).
ZnFe_2_O_4_ + hv → e^−^_CB_ + h^+^_VB_ + ZnFe_2_O_4_(1)
e^−^_CB_ + h^+^_VB_ → heat(recombination)(2)
h^+^_VB_ + Organic pollutants → [intermediate] *(3)
h^+^_VB_ + OH^−^ → OH(4)
Organic pollutants + OH → [intermediate] *(5)
[intermediate] * + OH → mineral salts + CO_2_↑ + H_2_O(6)
e^−^_CB_ + O_2_ → O_2_^−^(7)

A large group of studies have been conducted to enhance the optical properties of ZnFe_2_O_4_ nanomaterials because the photochemical activity of simple stacked of ZnFe_2_O_4_ is limited. ZnFe_2_O_4_ and its modified samples have been successfully prepared by hydrothermal processes, the coprecipitation technique, biosynthesis, and other methods. For example, ZnFe_2_O_4_, Zn_0.5_Cu_0.5_Fe_2_O_4_, ZnFe_1.9_Bi_0.1_O_4_, ZnFe_1.95_Ce_0.05_O_4_, Zn_0.5_Cu_0.5_Fe_1.9_Bi_0.1_O_4_ and Zn_0.5_Cu_0.5_Fe_1.95_Ce_0.05_O_4_ were synthesized by the solution combustion route for the photocatalytic removal of bisphenol A in wastewater [55]. ZnFe_2_O_4_/Fe_2_O_3_ hollow nanotubes were prepared for the degradation of ciprofloxacin [56].

Based on a literature search utilizing the ScienceDirect database with the keywords “zinc ferrite” and “photocatalysis”, 951 articles were found that had been published during last decade, 2012–2021, with most of the articles published in journals that are in the fields of Materials Science, Chemical Engineering, and Environmental Science (Figure 2a). Figure 2b shows the numbers of articles regarding different methods of preparing ZnFe_2_O_4_ and their variation over the last decades. In this paper, the basic properties of ZnFe_2_O_4_, the common preparation methods as well as their advantages and disadvantages and the research progress in modification were reviewed. By summarizing the current development and issues in ZnFe_2_O_4_ research, this mini-review aims to provides a reference for ZnFe_2_O_4_ photocatalytic technology in preparation and modification.

## 2. Synthesis of ZnFe_2_O_4_

The morphology, size, and structure of the photocatalyst particles can be affected by the synthesis method used, which in turn affects their catalytic performance, adsorption properties, etc. The bottom-up method (e.g., sol-gel, coprecipitation) generates nanoparticles by stacking atoms on top of each other and the resulting materials are homogeneous and free from impurities. The top-down method such as ball milling decomposes large particles of the photocatalytic material into small particles, but it has many limitations including impure products, uneven particles, crystal defects, uneven distribution, and surface structure defects.

ZnFe_2_O_4_ nanoparticles have been synthesized using grinding, combustion, ceramic, hydrothermal, solvothermal, co-precipitation, sol-gel, and biosynthesis methods. The following sections provide detailed descriptions of the morphologies and physical properties of ZnFe_2_O_4_ prepared using these methods.

### 2.1. Hydrothermal Synthesis/Solvothermal Process

The hydrothermal synthesis involves dissolving a 1:2 ratio of Zn^2+^ and Fe^3+^ in water, followed by the dropwise addition of ethylene glycol or ethanol with stirring to ensure that the solution is well-mixed. The resulting mixture is then placed into an autoclave and heated under high pressure to dissolve and recrystallize the precursor(s). The purity, size, saturation magnetization value, and photocatalytic activity of the product can be controlled by varying the concentration of the reactants, the hydrothermal temperature, reaction time, pH conditions, amount of surfactant, etc. In spite of its potential for the large-scale synthesis of ferrite, the hydrothermal synthesis has disadvantages including long reaction times, expensive autoclave equipment, and difficult to observe crystal growth. Because several materials used in hydrothermal synthesis are insoluble in water but readily soluble in organic solvents, the solvothermal method has been developed. The solvothermal method follows the same steps as hydrothermal synthesis with the exception that the precursor is dissolved in a non-aqueous organic solvent rather than water. In the solvothermal method, the solvent strongly affects the properties of the materials and the choice of an unsuitable solvent may even degrade the properties of the resulting materials. A schematic representation of the hydrothermal synthesis of ZnFe_2_O_4_ is shown in Figure 3.

Zhu et al. [57] prepared ZnFe_2_O_4_ by the hydrothermal method with the hydrothermal reaction temperature of 160, 180, and 200 °C, respectively. The averaged crystal size of ZnFe_2_O_4_ increased from 7.3 to 8.2 nm, the saturation magnetization value increased from 15.05 to 24.56 emu/g, and the crystal density increased upon increasing the hydrothermal temperature (Figure 4). It has also been found by Zhu et al. [57] that the magnetization value of the catalysts synthesized by hydrothermal synthesis was about six times that of the catalysts synthesized by the ceramic method, because part of the Fe^3+^ in the materials synthesized by hydrothermal synthesis occupied the A site and formed the complex spinel structure. A comparison by Lestari et al. [58] showed that the saturated magnetization value of ZnFe_2_O_4_ synthesized by the hydrothermal method was ~10 times higher than that of the sol-gel method.

Kurian et al. [59] synthesized ZnFe_2_O_4_ utilizing both a hydrothermal synthesis and solvothermal process using ethylene glycol and a small amount of polyethylene glycol as solvents. The hydrothermal method produced ZnFe_2_O_4_ with a particle size of 10.8 nm and crystallite size of 6.95 nm, whereas the solvothermal method produced a spherical porous structure composed of multiple 5–20 nm crystallites with a particle size of 87.3 nm. Not only did the nanoparticles prepared using the solvothermal method have a lower band gap of 0.3 eV than those prepared via the hydrothermal synthesis, but they also exhibited a saturation magnetization value of 57.76 emu/g, which was 7.6 times greater. Yoo et al. [60] found that the pure phase ZnFe_2_O_4_ was prepared by increasing in pH from 10 to 14 and reducing the hydrothermal reaction time from 12 to 5 h. Based on this, the filling pressure was controlled to obtain ZnFe_2_O_4_ nanoparticles of different diameters. When 70 mL of the reaction solutions was filled to the 100 mL reaction vessel, the average particle size was only 6 nm, which was much smaller than the critical size of 15–18 nm. At this size, the nanoparticles were transformed from the multi-domain to single-domain, which resulted in superparamagnetic behavior. Remanence was absent after the magnetic field was removed.

### 2.2. Co-Precipitation Technique

As for the co-precipitation technique, soluble metal ions (such as sulfate, nitrate, and chloride) are co-precipitated with a precipitant such as NaOH, oxalate, or carbonate in a solvent and the precipitate is then dried and sintered under an appropriate atmosphere and temperature to obtain crystalline oxides. The co-precipitation technique is suitable for large-scale production since it is simple and economic. However, there are disadvantages including the agglomeration of the final products, pH-dependent reaction, and dropwise addition of the precipitant. Figure 5 shows the general steps of the co-precipitation technique used for the synthesis of ZnFe_2_O_4_.

Nguyen et al. [61] showed that the purity of ZnFe_2_O_4_ obtained from the co-precipitation technique was significantly affected by the pH conditions; the higher the pH, the purer the product. At a pH of 6, α-Fe_2_O_3_ was produced due to insufficient Zn^2+^, and the precipitate was not pure. High purity ZnFe_2_O_4_ products were obtained upon increasing the pH to 13. ZnFe_2_O_4_ with particle diameters of 4 and 6 nm were prepared by Hoque et al. [38] by a co-precipitation technique. Using low-temperature spin kinetics and frequency dependence of the magnetization rate, it was demonstrated that ZnFe_2_O_4_ materials with these particle diameters are superparamagnetic.

### 2.3. Sol–Gel Method

The sol-gel method is a wet chemical method used to prepare ferrite nanoparticles. In this method, a solution containing precursors, usually metal chlorides and metal alkyl oxides, is first converted into gels by hydrolysis, condensation, and polymerization. The gel was further dried and calcined to finally obtain the desired crystalline oxides. The sol-gel method requires low reaction temperatures, no special equipment, and produces a uniform distribution of phases and particles in the multi-component system with controlled size and morphology, but the nanoparticles need to be heat treated to remove the impurities to improve the final purity of the materials. Figure 6 shows a schematic representation of the general steps used in the sol-gel synthesis of ZnFe_2_O_4_.

Somvanshi et al. [62] prepared ZnFe_2_O_4_ with a specific surface area of 11.534 m^2^/g using the sol-gel self-combustion method. Sonia et al. [63] found that ZnFe_2_O_4_ prepared using the sol-gel method had better homogeneity but slightly lower stability when compared to the material prepared via the co-precipitation technique. Prasad et al. [64] used citrate gel to synthesize ZnFe_2_O_4_ with an average size of 22 nm and saturation magnetization value of 12.87 emu/g. Kumar et al. [65] studied the effect of the calcination temperature on the material properties; an increase in the calcination temperature from 550 to 900 °C led to a decrease in the saturation magnetization value from 6.84 to 4.26 emu/g.

### 2.4. Biosynthesis Method

The features of simplicity, low cost, and low waste of biosynthesis make it an emerging synthetic method. It has been successfully demonstrated that nanoparticles comprised of metals and their oxides can be produced using alternative biological compounds such as the extracts from plant roots and leaves, fruit peels, seeds, biomass waste, sugar, and bacteria [66]. Vitamins, proteins, and mineral salts present in biomaterials provide nanoparticles with excellent compatibility and can also act as capping and reducing agents to enhance their stability and reduction rate [67]. Due to the biodiversity of plants, the preparation of nanoparticles via biosynthesis has a promising future.

Balasubramanian et al. [68] synthesized ZnFe_2_O_4_ via a biosynthesis method. The flower extract of nyctanthes arbor-tristis and a mixed solution of iron nitrate and zinc nitrate were thoroughly mixed and precipitated. The precipitate was then dried and calcined to obtain pure ZnFe_2_O_4_ nanoparticles. The uncalcined nanoparticles had a particle size of 4.5 nm, specific surface area of 200 m^2^/g, and pore size of 2.8 nm, which were 0.375, 4, and 2.92 times higher than those obtained using a calcination temperature of 800 °C, respectively. In the RhB degradation experiments, the ZnFe_2_O_4_ synthesized in this study was able to degrade 85–100% of the contaminant within 30 min. Surendra et al. [69] prepared a ZnFe_2_O_4_ nano sample with Jatropha extract assisted combustion, which had the morphology of porous flakes and agglomeration. Due to the small particle size and active sites in the starting material, which can remove up to 98% of malachite green under UV irradiation, it exhibited good photocatalytic activity. Rahmani et al. [70] dissolved metal nitrate precursors in a solution of Simbang Darah leaf extract. The hydrothermal synthesis was carried out and 10–80 nm spherical clusters of ZnFe_2_O_4_ were prepared, which were able to degrade 99.66% of direct red 81 dye after 2 h of solar irradiation. Tatarchuk et al. [71] used quince fruit extract to dissolve metal nitrate, where the obtained solution was heated for 30 min to evaporate. The formed gel ignited by itself and burned slowly with large amounts of gases. The resulting ferrite powder was ground and washed twice with distilled water to prepare paramagnetic Co_x_Zn_1−x_Fe_2_O_4_ (where x = 0.0 … 1.0 with step 0.2). The photodegradation rates of organic pollutants by ZnFe_2_O_4_ with different preparation methods are shown in Table 1.

### 2.5. Other Synthesis Methods

In addition to the above-mentioned methods, the methods used to prepare ZnFe_2_O_4_ also include microwave, ball milling, vapor deposition, acoustic chemical, pulsed laser, ceramic, and spray pyrolysis methods. Listed in Table 2 are the preparation principles, advantages, and disadvantages of these methods.

#### 2.5.1. Microwave-Assisted Method

The microwave-assisted method is a method that uses microwave to provide a thermally uniform nucleation environment for materials [79,80]. Microwave assisted synthesis technology has the characteristics of a fast heating rate, simple operation, short reaction time, high energy utilization efficiency, low cost, and high yield. However, this method requires complex equipment, which needs different reactors to apply different pressures, and needs long reactor operation times.

The ternary dual Z-scheme composite g-C_3_N_4_/ZnFe_2_O_4_/Bi_2_S_3_ was synthesized by Sarkar et al. [81] through a simple microwave-assisted process. Through the experiment, the best working condition was determined as the oxidant peroxymonosulfate (PMS) reached 1 g/L, the g-C_3_N_4_/ZnFe_2_O_4_/Bi_2_S_3_ (10 wt%) dose was 0.25 g/L, the concentration of 2,4,6-trichlorophenol (TCP) was 20 mg/L, and the degradation rate of TCP increased to 98.9% (Figure 7a–c). A dose of g-C_3_N_4_/ZnFe_2_O_4_/Bi_2_S_3_ (10 wt%) exceeding 0.25 g/L resulted in a reduced degradation rate of TCP due to catalyst particle agglomeration and increased solution turbidity. Increasing the pollutant concentration to 500 mg/L reduces the degradation efficiency to 60% due to the fact that, at larger TCP concentrations, more intermediates and by-products are produced, which consume free radicals and thus reduce the effective activity of the system (Figure 7d). In the reusability test, the TCP degradation rate decreased to 81.5% after five cycles, which was due to the leaching of Zn, Fe, and Bi ions from the catalyst after each cycle (Figure 7e).

Tamaddon et al. [82] prepared novel magnetically separable ZnFe_2_O_4_/methylcellulose spherical-shaped nanoparticles by microwave-assisted co-precipitation for the degradation of metronidazole (MNZ) under UV lamp irradiation. The surface morphologies of these spherical particles in the ZnFe_2_O_4_@methylcellulose were smooth, uniform, and compact, while the particles were loosely aggregated. The removal efficiency of MNZ was 92.65% in the synthetic samples and 71.12% in the real samples. The removal efficiency decreased with an increasing dose of the model pollutant due to the photocatalyst surface saturation at higher concentrations. Similar conclusions were also obtained by Sarkar et al. [81].

#### 2.5.2. Ball Milling Method

Ball milling is a process in which a mixture of metal salts is ground either dry or wet so that the compound is repeatedly broken and reconstituted. Ball milling has the advantages of low cost, simple operation, no use of toxic and expensive solutions, prevention of particle agglomeration, etc., and has the potential for large-scale production. The product purity of the ball milling method is affected by the grinding time. It takes tens of hundreds of hours to obtain a pure product, and there is a risk of material contamination during milling. Zhang et al. [83] obtained pure ZnFe_2_O_4_ by microwave-assisted ball milling for 30 h. The saturation magnetization value reached 82.23 emu/g, which was much higher than that obtained by high-energy ball milling (30 emu/g) [84], hydrothermal synthesis (12 emu/g) [85], and the solvothermal method (24.32 emu/g) [86].

#### 2.5.3. Sonochemical Method

The sonochemical method is a well-defined method for nanoparticle preparation that uses ultrasound as the driving force to accelerate the chemical reaction and increase product yield. When ultrasound is transmitted through a solution of precursors, the high energy of the sound ultrasound field is concentrated and causes acoustic cavitation in the solution. Accompanied by the collapse of the cavitation bubble, the energy is released in a very small space, so that the liquid environment produces an unusually high temperature to increase the speed of the chemical reaction. The method has the advantages of low energy consumption, high reaction rate, uniform product size, simple operation, and high economic benefit. However, there are also some disadvantages such as low magnetic moment and poor crystallinity. Patil et al. [87] prepared ZnFe_2_O_4_ with superparamagnetic behavior and a saturation magnetization of 5.321 emu/g by the sonochemical method. Raja et al. [88] synthesized ZnFe_2_O_4_/CeO_2_ nanocomposites by the sonochemical method for the degradation of methyl orange (MO) in visible light. Compared to the pure ZnFe_2_O_4_, the ZnFe_2_O_4_/CeO_2_ (1:1) nanocomposite exhibited a higher specific surface area and pore volume, thus providing more active sites for reactant molecules and reached a higher MO degradation efficiency.

## 3. Modification of ZnFe_2_O_4_-Based Photocatalysts

Since the band gap of ZnFe_2_O_4_ was only 1.9 eV, the photogenerated electron-hole pairs quickly recombined, thus decreasing its photocatalytic activity. To improve its photochemical performance, three ways of catalyst modification have been proposed: (1) Elemental doping, which introduces a second or even a third element; (2) composite modification, which compounds ZnFe_2_O_4_ nanoparticles with materials that have high activity and a low carrier recombination rate (e.g., metal oxides, metal sulfides, and carbon-based materials); and (3) morphological modification, which optimizes the nanostructure by turning the simple stacked low-dimensional materials into high-dimensional materials to increase the specific surface area, reaction sites, and carrier transport efficiency while decreasing the possibility of carrier recombination.

### 3.1. Elemental Doping

The photochemical properties of ZnFe_2_O_4_ can be enhanced upon doping into the gaps in the crystal structure of ZnFe_2_O_4_: Zn^2+^ at the A sites, Fe^3+^ at the B sites or O sites. Table 3 shows some of the metal ion doping for ZnFe_2_O_4_ and it is clear that the type, sites, and amount of doping affect the band gap and, in turn, the photocatalytic activity of ZnFe_2_O_4_. By using this modification method, it is possible to change the size of the cells in the material as well as increase the specific surface area and number of active sites. This can have a positive effect on its photochemical, electrical, and magnetic properties.

Musa et al. [95] prepared nanofibers comprised of pure ZnFe_2_O_4_ and Al-doped ZnFe_2_O_4_ (0.5% Al) using a combination of the electrospinning method and calcination. The diameter of the pure ZnFe_2_O_4_ nanofibers remained unchanged upon increasing the calcination temperature, but the diameter of the Al-doped ZnFe_2_O_4_ nanofibers decreased significantly, which proved that the Al ions had been successfully doped into the ZnFe_2_O_4_ crystal structure and changed the crystal structure and microscopic properties of the resulting samples (Figure 8a). On one hand, Al doping promoted the reversal of the cation distribution within the material and improved the magnetization value of the material, thus enhancing the recovery performance of the ZnFe_2_O_4_ photocatalyst; on the other hand, Al doping introduced an impurity energy level and inhibited the carrier complex. Al doping reduces the UV-Vis diffuse reflectance spectroscopy (DRS) absorbance of ZnFe_2_O_4_ (Figure 8b), and the energy band gap of the doped material was reduced to 1.7 eV (Figure 8c), and the percentage of RhB degradation increased from 25.66% to 38.83% after 60 min of visible light irradiation (Figure 8d).

Reyes-Rodríguez et al. [96] obtained Zn_1−x_Mg_x_Fe_2_O_4_ samples with an average diameter of 15.5 nm by doping Mg^2+^ into the ZnFe_2_O_4_ crystal structure using the sol-gel method. The saturation magnetization value of Zn_1−x_Mg_x_Fe_2_O_4_ first increased and then decreased with an increasing Mg^2+^ doping concentration, reaching a maximum value of 44.85 emu/g at x = 0.5. Son et al. [89] replaced Zn^2+^ with Ni^2+^ and the resulting Zn_1−x_Ni_x_Fe_2_O_4_ crystal structure was transformed into an inverse spinel structure. The oxygen vacancies increased, and the material band gap was reduced to a minimum value of 1.72 eV with an increasing Ni^2+^ doping ratio, thus improving the photocatalytic performance.

Peymani-Motlagh et al. [97] prepared Co_0.5_Zn_0.5_Fe_1.9_R_0.1_O_4_ (R = Gd, Pr, Sm) upon introducing Gd^3+^, Pr^3+^, and Sm^3+^ into the B sites, and the photocatalytic activities of the observed MO degradation were in the order of: Gd^3+^ > Pr^3+^ > Sm^3+^. Borhan et al. [98] prepared ZnFe_2−x_Cr_x_O_4_ (x = 0, 0.25, 0.5, 1.0, 1.5, 2.0) using the sol-gel method. The saturation magnetization value decreased from 4.35 to 3.25 emu/g upon increasing the chromium content.

Li et al. [99] prepared a carbon-nitrogen doped ZnFe_2_O_4_ yolk-shell structure by adding dopamine as a carbon and nitrogen source. The carbon and nitrogen ions replaced the oxygen anions, and the energy band position was slightly negatively shifted, which expanded the absorption range of visible light. The internal void of the yolk-shell structure could produce a stronger scattering of incident light. The internal void of the yolk-shell structure could also produce a stronger scattering of incident light. The degradation rate of the yolk-shell structured ZnFe_2_O_4_ was 1.73 times higher than that of undoped ZnFe_2_O_4_. The yolk-shell structured ZnFe_2_O_4_, after doping with C and N, exhibited a 1.3-fold higher photocatalytic performance than that of the broken structure after doping with C and N.

### 3.2. Composite Modification

Since 46% of the complete energy in sunlight is situated in the visible light region, ZnFe_2_O_4_ nanoparticles with narrow band gaps (1.9  eV) could be more adequate than traditional semiconductors such as TiO_2_ (3.2  eV) and ZnO (3.2  eV) in employing solar energy to photocatalytically degrade contaminants. However, few semiconductor photocatalysts with their band edge potentials are suitable on their own. Combining ZnFe_2_O_4_ with other semiconductors, metals, and carbon materials is necessary for effective catalyst applications. The key requirement for new photocatalysts is to optimize the band gap so that the electron complexation rate is negligible and to make the CB edge more negative and the VB edge more positive. That is, to make the semiconductor highly reducible for electrons in the CB (+0.5 to −1.5 V vs. normal hydrogen electrode (NHE)) and highly oxidizable for holes in the VB (+1.0 to +3.5 V vs. NHE) (Figure 9).

#### 3.2.1. Composites of ZnFe_2_O_4_ and Conducting Polymers

Commonly used conducting polymers include polyacetylene (PCE), polypyrrole (PPY), polyindole (PBE), and polyaniline (PANI), and their copolymers, which are simple and inexpensive to prepare. The combination of ZnFe_2_O_4_ with conducting polymers not only reduced the material agglomeration, but also increased the number of active sites, shortened the carrier diffusion pathway, and improved the carrier mobility.

Using the in situ polymerization method, Das et al. [101] prepared a series of polypyrrole (PPY) sensitized zinc ferrite/graphitic carbon nitride (ZFCN) heterojunction (ZFCN@10PPY, ZFCN@20PPY, and ZFCN@30PPY) nanocomposites. High resolution transmission electron microscope images showed that the edges of the ZnFe_2_O_4_ and g-C_3_N_4_ (graphitic carbon nitride, CN) lattice were interconnected (Figure 10a), indicating that the two components were not simply deposited, but formed a heterogeneous structure. The nitrogen adsorption-desorption isotherms and pore size distribution, as shown in Figure 10b,c, indicate that the specific surface area of ZFCN@20PPY was higher when compared with the CN, PPY, and ZFCN, and the pores that formed were narrower. Figure 10d shows that ZFCN@20PPY had a very low luminescence spectral intensity, confirming that the low band gap polypyrrole can act as a photosensitizer to provide additional electrons for ZnFe_2_O_4_/CN. The photocatalytic mechanism of the material is shown in Figure 10e, in which the photoexcited electrons could easily migrate from PPY to the CB of CN, and then further transfer to the CB of ZnFe_2_O_4_, and the photogenerated electrons captured reactive oxygen species such as superoxide and hydroxyl groups on the ZnFe_2_O_4_ surface to degrade the pollutants.

Zhang et al. [102] successfully prepared a dual Z-scheme heterojunction photocatalyst PANI/BiOBr/ZnFe_2_O_4_ (Figure 11a) using the hydrothermal method. The lone pairs of electrons on the outer nitrogen atoms of PANI can enhance the adsorption of pollutant molecules and the BiOBr flower-like structure improves the oxidation ability of the composite. By forming heterojunctions, the visible response range is widened to include the entire visible spectroscopy. A slight blue shift was observed as the irradiation time increased, caused by de-ethylation and the breakdown of the conjugate structure of RhB as the intensity of the characteristic peak at 554 nm decreased (Figure 11b). The reaction solution also turned from magenta to colorless. The results showed that the PANI/BiOBr/ZnFe_2_O_4_ composites exhibited a higher stability than that in RhB degradation (Figure 11c). When exposed to visible light, pure BiOBr showed a degradation rate of 50.26%, compared to the previous of 60.25%. However, RhB degradation by the PANI/BiOBr/ZnFe_2_O_4_ composite material was not significantly affected, decreasing from 99.26% to 97.86%. As shown in Figure 11d, a dual Z-scheme reaction mechanism based on the results of fluorescence photoluminescence (PL) spectroscopy, free radical capture experiments, ESR spin-trap with DMPO technique, calculation using the horizontal axis intercept of the Mott-Schottky plots has been proposed. PANI (−0.49 V vs. NHE) and ZnFe_2_O_4_ (−0.90 V vs. NHE) accumulate photogenerated electrons with strong reducing abilities, and these electrons can easily reduce O_2_ to O_2_^−^. In addition to directly oxidizing pollutants, photogenic holes (3.08 V vs. NHE) that accumulated in the VB of BiOBr can also oxidize H_2_O/OH^−^ to radical OH.

#### 3.2.2. ZnFe_2_O_4_ Composite Using Metal Oxides

Combining ZnFe_2_O_4_ with some metal oxides with strong oxidation ability, high carrier separation efficiency, and an ability to inhibit electron-hole recombination can improve its performance. Wang et al. [103] prepared S-type heterojunction ZnFe_2_O_4_/SnFe_2_O_4_ photocatalysts using a one-step solvothermal method. The experimental results showed that a moderate amount of SnFe_2_O_4_ (5%) promoted carrier separation and enabled the material to achieve the best visible light absorption efficiency, but an excess of SnFe_2_O_4_ led to a wider band gap and hindered the photogenerated carrier capability (Figure 12a). The 5% SnFe_2_O_4_/ZnFe_2_O_4_ PL intensity was the lowest in all materials (Figure 12b), transient photocurrent was 17.5 × 10^−8^ A, about pure ZnFe_2_O_4_ (10 × 10^−8^ A), and 1.75 times that of pure SnFe_2_O_4_ (1 × 10^−8^ A) (Figure 12c). It was further proved that an appropriate amount of SnFe_2_O_4_ was favorable for carrier separation and charge transfer behavior in nano-heterojunction materials. The SnFe_2_O_4_ composite doubled the saturation magnetization strength of ZnFe_2_O_4_ and enabled the composite to be separated from the aqueous solution quickly and easily (Figure 12d).

Al-Shwaiman et al. [104] prepared CeO_2_/ZnFe_2_O_4_ using a simple chemical co-precipitation technique. The specific surface area of CeO_2_/ZnFe_2_O_4_ was 126.99 m^2^/g and the mesopore pore size was 2.96 nm, which was 1.46 and 1.16 times that of pure ZnFe_2_O_4_ and 1.31 and 1.50 times that of CeO_2_, respectively. During the degradation of MB, the photocatalytic efficiency of the composites was 1.57 times that of pure ZnFe_2_O_4_ and 2.75 times that of CeO_2_, which verified the improvement in the photocatalytic efficiency observed upon increasing the specific surface area and mesopore size.

#### 3.2.3. ZnFe_2_O_4_ Composite Using Metal Sulfides

The presence of elemental sulfur promotes the electron transfer rate of metal sulfides in redox reactions, which makes the conductivity of metal sulfides better than that of the metal oxides. In addition, metal sulfides also have the advantages of good mechanical stability, high thermal stability, easy synthesis, and low cost. It is important to note that metal sulfides exhibit serious photocorrosion, photochemical instability, and other disadvantages, which greatly affect their stability. The combination of metal sulfides with ZnFe_2_O_4_ can induce the separation of the photogenerated carriers and inhibit their recombination, which improves the photochemical performance of ZnFe_2_O_4_.

Akshhayya et al. [105] prepared ZnFe_2_O_4_/SnS_2_ using an acoustochemically-assisted method and the synergistic effect of the materials resulted in a lower PL spectral intensity in the composites when compared to ZnFe_2_O_4_ and SnS_2_, a lower electron-hole pair recombination rate, and the introduction of defects in the lattice structure of the composites, which increased the transient photocurrent and further improved the electron mobility. Singh et al. [106] prepared CdS-modified ZnFe_2_O_4_ nanocubes using a hydrothermal synthesis. Modification using CdS nanoparticles significantly reduced the PL intensity and improved the electron-hole separation. The nanocubes comprised of the ZnFe_2_O_4_ and CdS nanoparticles exhibited a synergistic effect to widen the PL band gap and improved the UV-Visible absorption range.

#### 3.2.4. ZnFe_2_O_4_ Composites Using Carbon-Based Materials

Carbon materials such as graphene and graphitic carbon nitride have the advantages of a large specific surface area, good dispersion, strong adsorption ability, high electron mobility, and stable performance. Different kinds of carbon materials combined with ZnFe_2_O_4_ can effectively improve the photochemical properties of ZnFe_2_O_4_ [107].

Jun et al. [108] prepared a unique poly hybrid by depositing lanthanum oxide carbonate (La_2_CO_5_) and ZnFe_2_O_4_ on reduced graphene oxide (LaZF@rGO) using a hydrothermal-precipitation method. Compared with the pure ZnFe_2_O_4_, the band gap of the composite was reduced by 0.44 eV (Figure 13a). The activities of the photocatalytic materials in different proportions were arranged as follows: 1:1 LaZF@rGO (0.0612 min^−1^) > ZF@rGO (0.0314 min^−1^) > 2:1 LaZF@rGO (0.0218 min^−1^) > 4:1 LaZF@rGO (0.0105 min^−1^) > 5:1 LaZF@rGO (0.0074 min^−1^) (Figure 13b). Figure 13c shows the inhibition effect of different free radical scavengers on the reaction, and the inhibition effect of benzoquinone (BQ) was the most obvious. It can be seen that the addition of the catalyst can generate superoxide radicals under light. The magnetization value of LaZF@rGO was 24.59 emu/g, which was lower than that of ZnFe_2_O_4_ (41.90 emu/g), but still sufficient for its magnetic separation from aqueous solutions (Figure 13d). As shown in Figure 13e, the electrons excited by La_2_CO_5_ at the CB were transferred to the CB of the ZnFe_2_O_4_ through the rGO surface. The VB holes of La_2_CO_5_ migrate to the VB of ZnFe_2_O_4_ with the help of rGO. The rGO acts as a medium for electron-hole transport, thereby inhibiting the recombination of electron-hole pairs.

Baynosa et al. [109] prepared ZnFe_2_O_4_/rGO using a one-step hydrothermal synthesis and found that the interaction between the ZnFe_2_O_4_ nanoparticles and graphene flakes limited the crystallization of ZnFe_2_O_4_, so the addition of acetate acts as a protective agent to promote the stable generation of nanocrystals. The specific surface area of the ZnFe_2_O_4_/rGO composite after loading the nanoparticles onto graphene flakes was as high as 141.3 m^2^/g, which was 1.67-flod higher than the surface area of the rGO-ZnFe_2_O_4_ hybrid prepared using a hydrothermal synthesis reported by Wang et al. [110]. Gao et al. [111] synthesized C-ZnFe_2_O_4_ by the microwave induction method, which could absorb 90% of the incident waves and had good pumping properties.

### 3.3. Morphological Modification

As the morphology of ZnFe_2_O_4_ change, its carrier transport efficiency, surface area, active sites, and light absorption capacity change, which affect its photochemical properties. Recently, many researchers have explored the different morphologies of ZnFe_2_O_4_ and their photochemical properties. According to the extent to which the spatial dimension of the nanomaterials is constrained by the nanometer size, ZnFe_2_O_4_ can be classified as zero-, one-, two-, or three-dimensional. Zero-dimensional nanomaterials mainly include nanoparticles and nanoclusters, which are the basis of all morphological modifications and usually have advantages such as a large specific surface area and abundant active sites, but also have disadvantages such as facile agglomeration and poor light absorption. The current modifications of zero-dimensional ZnFe_2_O_4_ are mainly divided into two types: (1) Adjusting the microstructure of the ZnFe_2_O_4_ nanoparticles by adjusting the preparation conditions such as solvent, temperature, precipitant, and pH; and (2) assembling ZnFe_2_O_4_ onto other high-performance or high-dimensional photocatalysts to improve the photocatalytic performance. Figure 14 shows the SEM images of ZnFe_2_O_4_ with different morphologies.

#### 3.3.1. One-Dimensional ZnFe_2_O_4_

One-dimensional nanomaterials include nanowires, nanotubes, and nanorods. The wire structure can reduce the agglomeration of spherical particles, expose more active sites, and improve the radial electron transport efficiency. Ghazkoob et al. [112] prepared ZnFe_2_O_4_ nanowires and ZnFe_2_O_4_ nanowire/BiVO_4_ composites with an average length of 6.3 μm and an average diameter of 38 nm using AC pulse electrodeposition and hydrothermal synthesis, respectively. The nanowire structure increased the contact area between the catalyst and reaction solution and improved the photocatalytic efficiency, but the magnetic nanowires of a single material aggregated and produced curling, and the adjacent magnetic moments were not parallel in the curling mode, which led to the reduction in the nanowire saturation magnetization value. The BiVO_4_ coating effectively reduced the aggregation and curling among the ZnFe_2_O_4_ nanowires and improved the magnetization value of the material.

Zhang et al. [116] synthesized nanotubes composed of ZnFe_2_O_4_ nanotubes with a diameters of 5~10 nm using carbon nanotubes as a template (Figure 15a). The nanotube structure not only increased the specific surface area, but also the saturation magnetization value up to 45.4 emu/g. Liu et al. [113] developed a method to fabricate hollow oxides on electrospun nanofiber templates using the atomic layer deposition of oxides, and the resulting products not only retained the nanofiber structure between the tubes, but also appeared to have hollow structures in their bodies (Figure 15b). The extra-long one-dimensional nanofibers provided good radial transport channels for carriers and increased the ability of charge trapping by the catalyst surface, thus facilitating the redox reaction. The hollow structure shortened the carrier transport distance from the interior to the surface, increased the amount of light scattering within the fiber, and improved the light absorption capacity.

#### 3.3.2. Two-Dimensional ZnFe_2_O_4_

Two-dimensional nanomaterials are only available as nanofilms with a high specific surface area, low flat-band dot positions, large charge layer widths, and short carrier diffusion lengths, which make them ideal for developing photocatalysts. Zhu et al. [117] prepared new porous ultrathin ZnFe_2_O_4_ nanofilms with a different number of layers using the Langmuir–Blodgett (LB) method on ZnFe_2_O_4_ that was close to the monolayer, closely aligned at a surface pressure of 40 mN/m, and a deposition rate of 1 mm/min. Figure 16a,b shows that the single layer ZnFe_2_O_4_ (LB-ZFO-1) had the highest corresponding SPV signal of 4.87 μV compared to the multi-layer ZnFe_2_O_4_. The quantum efficiency of the single layer was 1.7 times that of the five layer (LB-ZFO-5) and 2.6 times that of the nine layer (LB-ZFO-9). It was proven that the monolayer ZnFe_2_O_4_ thin film had better efficiency of the photogenerated carrier separation and charge transfer behavior. Both monolayer and multilayer ZnFe_2_O_4_ films in the 500, 1000 and 2500 Hz frequency measurements of the Mott–Schottky figure showed a positive slope of the linear area, indicating that the preparation of the ZnFe_2_O_4_ film was indeed an n-type semiconductor (Figure 16c). Comparing the capacitive properties of the TiO_2_ porous films with ZnFe_2_O_4_, they found that both the monolayer and multilayer films had capacitive properties, which led to the degradation of the photocatalytic properties of the films (Figure 16d).

Chen et al. [118] prepared translucent ZnFe_2_O_4_ films on MgAl_2_O_4_ substrates using pulsed laser deposition with a Curie temperature close to 327 °C. Luo et al. [119] prepared ZnFe_2_O_4_/Fe^3+^-TiO_2_ heterojunction composite films using lithography-assisted chemical solution deposition techniques. The film with a 5 μm thickness had a surface contact angle as low as 3° in the MB solution and was super-hydrophilic. With the recycling of the material, the film was gradually soaked, and the time to degrade the pollutants of the same concentration decreased from 24 min to 19 min.

Erusappan et al. [120] synthesized ZnFe_2_O_4_/BN two-dimensional heterostructure nanocomposites with different wt% of boron nitride (BN) by the solvothermal method. The incorporation of 9.3% BN over the ZnFe_2_O_4_ catalysts showed the best synergistic effect between Zn and BN, which enhanced the charge separation and electron transport efficiency. The degradation rate of dyes (CR) (Figure 17a) and antibiotics (Figure 17b) reached 0.00982 and 0.01019 min^−1^, respectively, showing the best proof. Almost no activity loss was found (Figure 17c,d) after four cycles of reuse, indicating good photostability.

#### 3.3.3. Three-Dimensional ZnFe_2_O_4_

There are a wide variety of three-dimensional nanomaterials, which are usually low-dimensional nanomaterials that have been modified or combined to form three-dimensional composite nanomaterials such as nanowire arrays [121], hollow spheres [122], and frameworks [123,124]. Three-dimensional nanomaterials can effectively use the physical structure to expose more active sites and improve light absorption.

Chen et al. [114] constructed a NiS-modified CuO@ZnFe_2_O_4_ hierarchical nanoarray structure photocatalyst that was composed of CuO nanowires as the core and ultrathin ZnFe_2_O_4_ nanosheets as the shell, as shown in Figure 18. The nanowire array structure not only had large gap channels, a large specific surface area, and many active sites, but also had a high incident light capture capability. Meanwhile, the layered structure allowed the charge to be transferred outward from the inner part of the catalyst layer-by-layer, which significantly reduced the charge complexation rate.

Hollow structures such as hollow spheres and hollow skeletons have the advantages of low density, good dispersion, large specific surface area, short carrier transfer pathway, adjustable photorefractive index, strong light scattering ability, and high plasticity, which make them a promising research direction. Hollow structures are synthesized using hard-template, soft-template, and self-template methods.

Zhang et al. [125] synthesized double-shell Ag/AgCl/polydopamine (PDA)/ZnFe_2_O_4_ nanocubes using hollow ZnFe_2_O_4_ nanospheres (G-ZnFe_2_O_4_) prepared via the hydrothermal and self-etching process with PDA used as the template and reducing agent. The PDA template was dissolved in KOH solution to form double-shelled Ag/AgCl/ZnFe_2_O_4_ hollow nanocubes (DAGZNs). Due to the dissolution of the PDA, the specific surface area of DAGZNs increased from 43.90 (before the removal of the template) to 79.73 m^2^/g, and the pore size expanded from 2.69 to 6.84 nm (Figure 19a,b). Figure 19c indicates that the PL intensity of the DAGZN photocatalyst was significantly higher than that of the bulk material, and that its high photocatalytic activity may be due to its double-shell hollow structure. The transient photocurrent in Figure 19d shows that the photocurrent response of DAGZNs was stronger than that of the Ag/AgCl/PDA/G-ZnFe_2_O_4_ nanoparticles, which further demonstrates the advantage of the hollow multi-shell structure. Figure 19e shows a schematic representation of the light scattering effect in the hollow double-shell structure and the possible response mechanism of the photocatalytic performance of DAGZNs under visible light irradiation proposed by the authors based on their results. The external cubic structure and internal hollow sphere structure increased the number of channels for light scattering inside the material, and these channels could effectively enhance the light absorption, and therefore the photocatalytic activity.

Cao et al. [115] synthesized a novel hollow porous cube ZnFe_2_O_4_ via calcination using a metal-organic skeleton (Prussian blue (PB)) as a precursor. During calcination, the organic matter in the precursor slowly decomposed, in which the gas diffused outward to create pores in the ZnFe_2_O_4_ oxide layer. The structure could increase the amount of light scattered and improve the photocatalytic activity.

### 3.4. Pollutant Degradation Performance of Photocatalysts Prepared Using Different Modification Methods

ZnFe_2_O_4_ has a narrow band gap of 1.9 eV, which allows the material to easily generate electron-hole pairs that sequentially produce oxygen-containing substances. Therefore, ZnFe_2_O_4_ nanoparticles are widely used in the study of the removal of organic pollutants such as organic dyes, phenols, antibiotics, etc.

#### 3.4.1. Organic Dyes

Organic dyes are typical pollutants in wastewater from the textile industry, which is one major source of industrial wastewater pollution [126]. Organic dyes can be divided into cationic dyes (RhB, MB), anionic dyes (methylene orange (MO), (Congo red (CR)), and disperse dyes according to the ionic charge they dissociated into in the aqueous medium. In the photocatalytic degradation of organic dyes, the dye molecules are first adsorbed on the surface of the photocatalyst and then participate in the redox reaction. Chahar et al. [127] fabricated Co_x_Zn_1−x_Fe_2_O_4_ (x = 0.0, 0.1, 0.2, 0.3, 0.4 and 0.5) for MB degradation. The results showed that Co_0.5_Zn_0.5_Fe_2_O_4_ had the highest MB degradation efficiency (77%), which was distinctly higher than that of the undoped ZnFe_2_O_4_. Shakil [128] prepared (Cd_x_Zn_1−x_Co_0.25_Fe_1.75_O_4_; x = 0, 0.25, 0.5 and 0.75) for the removal of MO under visible light. Based on the PL spectroscopy analysis, the substitution of Co and Cd resulted in a decrease in PL intensity, which led to reduced photogenerated electron–hole pair recombination thus increases the photocatalytic efficiency. Meanwhile, the composite also showed good reusability and stability.

#### 3.4.2. Phenols

Phenolic compounds are widely found in various industrial processes such as oil refining, the petrochemical industry, the pharmaceutical industry, and resin manufacturing [129]. Absorption combined with incineration is one of the most common method of removing phenolic compounds from aqueous solutions. However, the incineration of phenolic compounds will emit gaseous pollutants such as dioxins, which effect both the environment and human health [130]. Photocatalytic degradation of phenolic compounds is a much more environmentally friendly technique. Mady et al. [131] fabricated aγ-MnO_2_@ZnFe_2_O_4_/rGO nanohybrids for phenol degradation using a one-pot hydrothermal self-assembly method. The nanohybrid had a specific surface area as high as 376.88 m^2^/g, which provided a large number of adsorption/active sites. The catalytic efficiency of the γ-MnO_2_@ZnFe_2_O_4_/rGO nanohybrid was the highest (100%) compared to the other composites including δ-MnO_2_@ZnFe_2_O_4_, γ-MnO_2_@rGO, and ZnFe_2_O_4_@rGO. This high efficiency may be attributed to the high surface area of the nanohybrid compared to the other composites and the ability of both MnO_2_ and ZnFe_2_O_4_ to activate PMS through an electron transfer mechanism to produce sulfate radicals.

#### 3.4.3. Antibiotics

Antibiotics usually come from pharmaceutical industries, which produce pharmaceuticals and personal care products. The direct discharge of wastewater containing untreated antibiotics will endanger human and animal health by bio-accumulating and bio-magnifying into the food chain, increasing antimicrobial resistance and inhibiting the growth of bacteria that are beneficial to the eco-system [132]. Dhiman et al. [133] synthesized Ni_1−x_Zn_x_Fe_2_O_4_ (x = 0, 0.1, 0.3, and 0.5) for levofloxacin degradation using the solution combustion method. Ni_0.1_Zn_0.9_Fe_2_O_4_ showed the highest degradation efficiency of levofloxacin (96.8%) within 90 min under visible light irradiation. O^2−^ was regarded as the main active species in the photocatalysis reaction. Levofloxacin was degraded into low molecular weight compounds through four pathways including de-piperazinylation, decarboxylation, quinolone moiety transfer, and piperazine ring oxidation.

Table 4 lists some of the results obtained in recent studies on the photocatalytic performance of ZnFe_2_O_4_-based composites, which are prepared by various modification methods in pollutant degradation. It can be seen that most of the ZnFe_2_O_4_-based composites showed high photocatalytic activity within a short reaction time. A few studies have also shown a good degradation effect on actual wastewater such as pharmaceutical wastewater.

## 4. Conclusions

ZnFe_2_O_4_ is a photocatalyst that exhibits high stability, high carrier separation efficiency, and easy recovery due to its spinel structure, narrow band gap, and special magnetic properties. Commonly used ZnFe_2_O_4_ synthesis methods include the hydrothermal/solvothermal method as well as the co-precipitation, sol-gel, and biosynthesis methods. Recently, various catalyst modification methods have been studied such as elemental doping, composite modification, and morphological modification to improve the photocatalytic activity and reusability of ZnFe_2_O_4_. Great progress has been made in ZnFe_2_O_4_ photocatalyst research, but many challenges still remain. Therefore, it is necessary to conduct further research on ZnFe_2_O_4_ in the field of photocatalysis, focusing on the following aspects:(1)Developing innovative preparation methods. Hydrothermal, co-precipitation, and sol–gel methods are relatively mature, but they are expensive and energy-intensive. Expanding the biosynthesis concept to each preparation method and the development of green methods could reduce the particle size of the nanoparticles and improve their active sites at the same time, with low cost, simple operation, and no chemical pollution. It is possible to improve the biosynthesis methods by studying trace minerals or enzymes, and further research into the synthesis mechanisms can be conducted based on the enzymes and minerals present in plants.(2)Expanding materials in a modified way. In order to better suit the practical applications, a variety of methods such as doping and composite modification may be used to improve the performance and stability of single ZnFe_2_O_4_ photocatalysts. By doping different elements with defects or combining them with strong redox-capable nanomaterials, heterojunctions can be formed to adjust the material band gap, improve carrier separation, and reduce carrier recombination, thereby improving the photochemical performance.(3)There is a need for further investigation into the light absorption mechanisms of different morphologies. At present, the research on the morphological modification of ZnFe_2_O_4_ is relatively limited, which mainly focuses on one-dimensional modification. Some studies have shown that the selection of different dimensional modifications leads to an exponential increase in the light absorption capacity. An in-depth understanding of the light absorption and photocatalytic mechanisms of multidimensional materials is needed. Research can be conducted on a number of excellent three-dimensional structures such as hollow multi-shell structures, nanosheet arrays, etc.

ZnFe_2_O_4_ photocatalysts have received increasing research attention in recent years. With the efforts of researchers and the advancement of preparation and characterization techniques, it is believed that ZnFe_2_O_4_ photocatalysts can be widely utilized in industrial applications in the near future.

## Figures and Tables

**Figure 1 ijerph-19-10710-f001:**
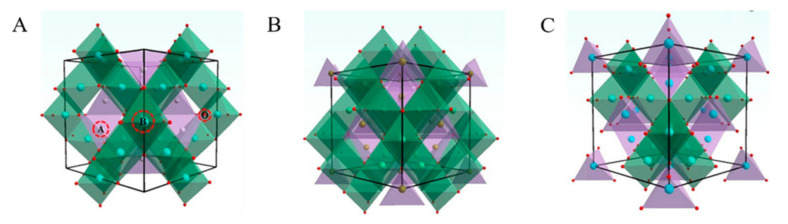
The crystal structure of spinel with different structures: (**A**) normal spinel; (**B**) inverse spinel; (**C**) complex spinel [31] (reprinted with permission from [31] Zhao et al., 2017).

**Figure 2 ijerph-19-10710-f002:**
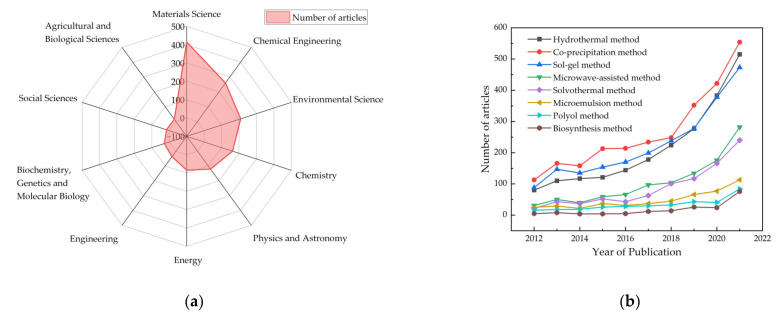
The literature survey on ZnFe_2_O_4_ from 2012 to 2021: (**a**) fields of the relevant publications; (**b**) degree of attention to the different preparation methods used.

**Figure 3 ijerph-19-10710-f003:**
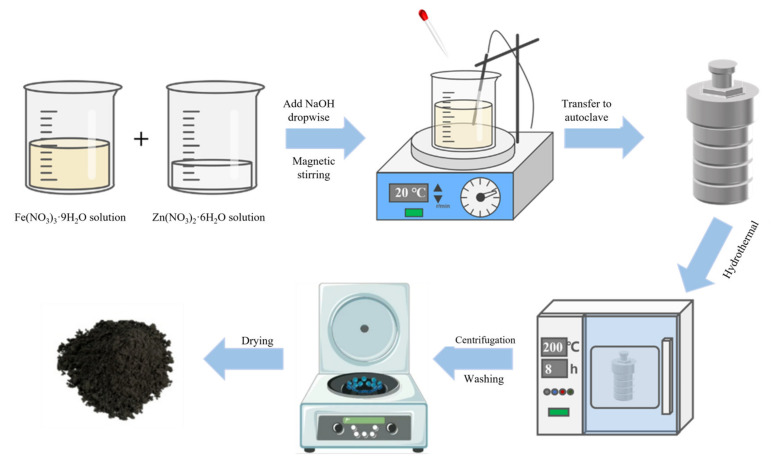
A schematic representation of the general steps used in the hydrothermal synthesis of ZnFe_2_O_4_.

**Figure 4 ijerph-19-10710-f004:**
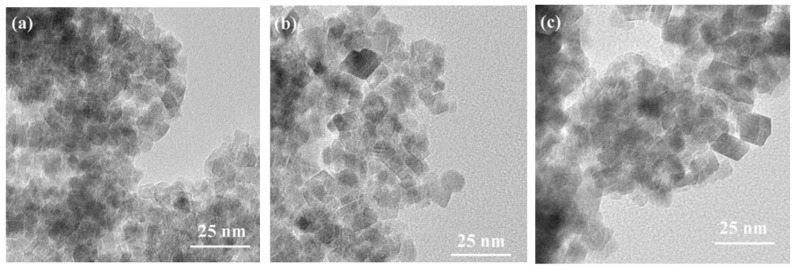
The transmission electron microscope images of the ZnFe_2_O_4_ nanoparticles prepared at different hydrothermal temperatures: (**a**) 160; (**b**) 180; and (**c**) 200 °C [57] (reprinted with permission from [57] Zhu et al., 2021).

**Figure 5 ijerph-19-10710-f005:**
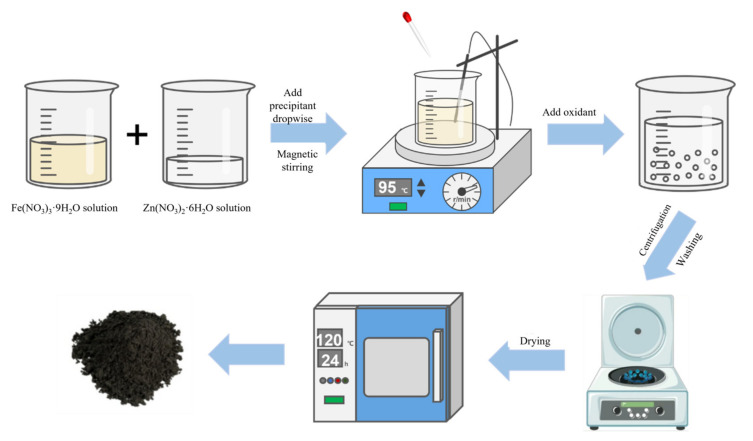
A schematic representation of the general steps used for the co-precipitation technique of ZnFe_2_O_4_.

**Figure 6 ijerph-19-10710-f006:**
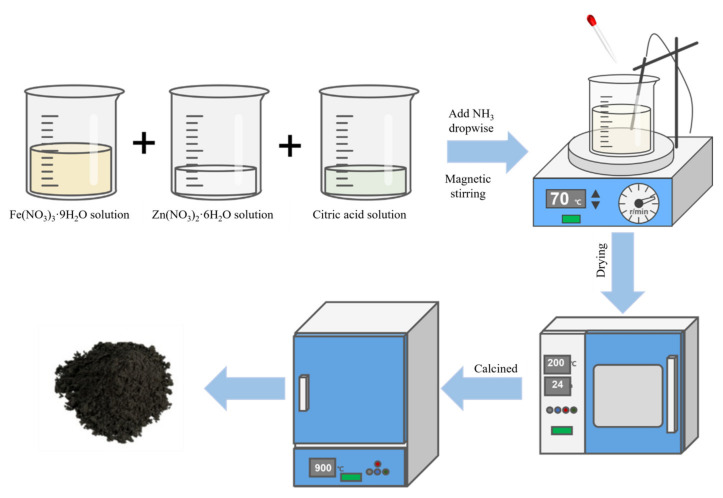
A schematic representation of the general steps used for the sol-gel synthesis of ZnFe_2_O_4_.

**Figure 7 ijerph-19-10710-f007:**
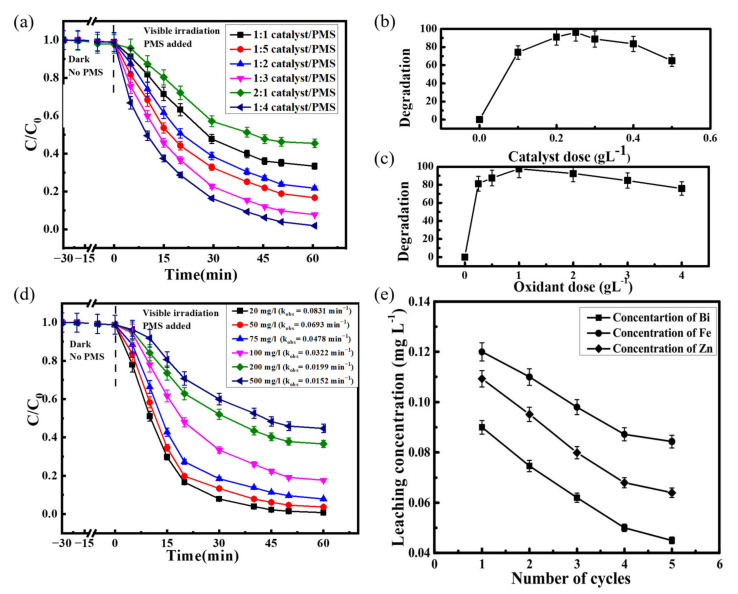
(**a**) The TCP degradation using different g-C_3_N_4_/ZnFe_2_O_4_/Bi_2_S_3_ catalysts: PMS dosage; (**b**) determination of the optimum catalyst dosage; (**c**) determination of the optimum PMS dosage. Reaction conditions: [TCP] = 50 mg/L, Temperature [T] = 35 °C, visible light intensity: 80 W; (**d**) variation of residual TCP concentration for various initial concentration; (**e**) leached ion concentrations after consecutive cycles [81] (reprinted with permission from [81] Sarkar et al., 2022).

**Figure 8 ijerph-19-10710-f008:**
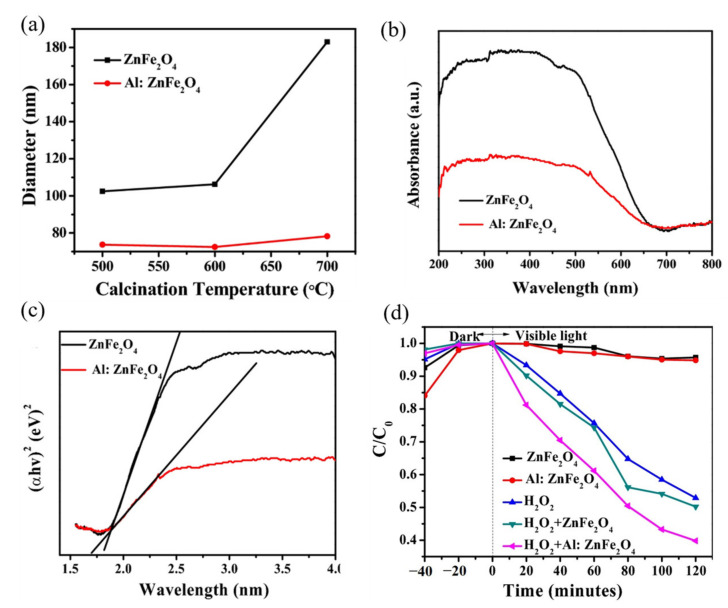
The ZnFe_2_O_4_ nanofibers and Al-doped ZnFe_2_O_4_ nanofibers: (**a**) plots of fiber diameters vs. calcination temperature; (**b**) UV-Vis DRS; (**c**) Tauc’s plots of calcined at 600 °C; (**d**) photocatalytic degradation rates of RhB by the pure and Al-doped ZnFe_2_O_4_ and their photo-Fenton process, with an H_2_O_2_ oxidant [95] (reprinted with permission from [95] Musa et al., 2022).

**Figure 9 ijerph-19-10710-f009:**
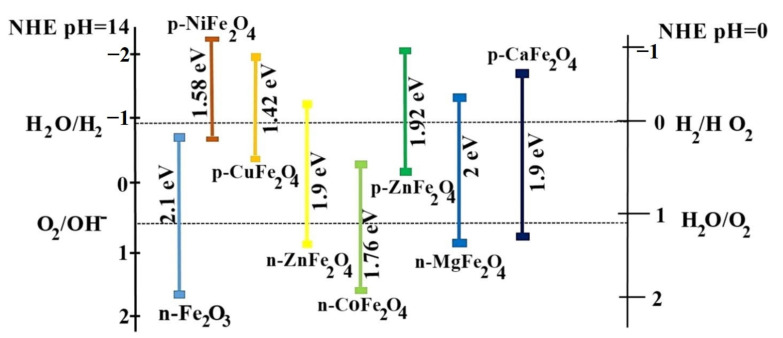
The band gap positioning of spinel ferrites upon interaction with an aqueous solution referenced with NHE (right pH ¼ 14 and left pH ¼ 0) relative to the standard potentials for the reduction and oxidation of water [100] (reprinted with permission from [100] Sonu et al., 2019).

**Figure 10 ijerph-19-10710-f010:**
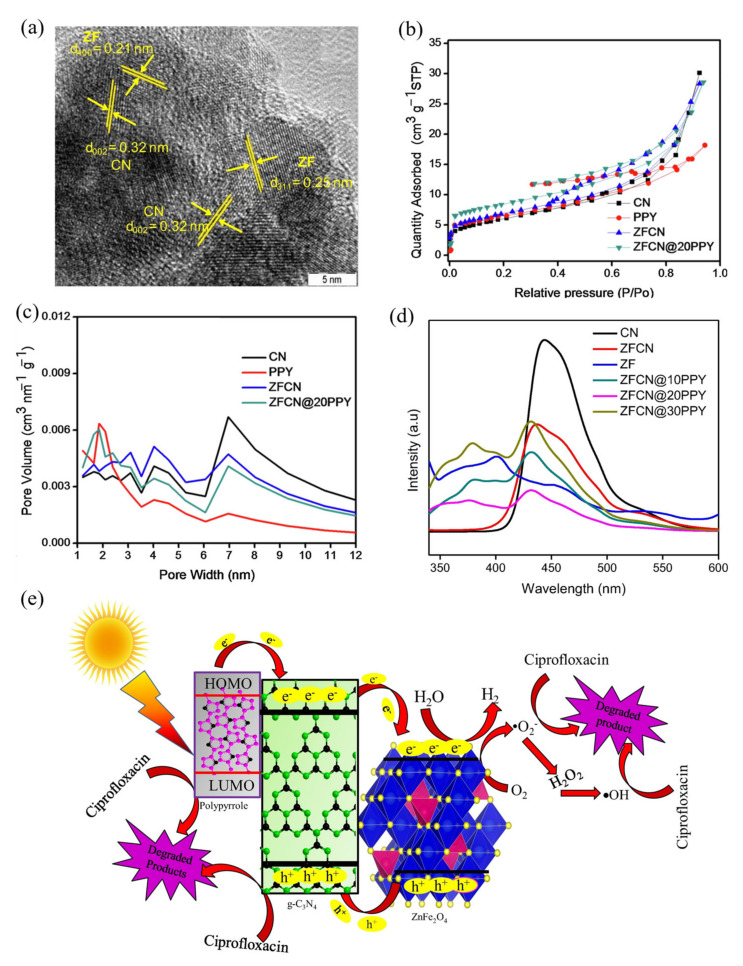
(**a**) High resolution transmission electron microscope images of ZFCN@20PPY; (**b**) N_2_ adsorption-desorption isotherms obtained for PPY, CN, ZFCN, and ZFCN@20PPY; (**c**) pore size distribution; (**d**) photoluminescence (PL) spectroscopy; (**e**) a schematic representation of the reaction mechanism [101] (reprinted with permission from [101] Das et al., 2020).

**Figure 11 ijerph-19-10710-f011:**
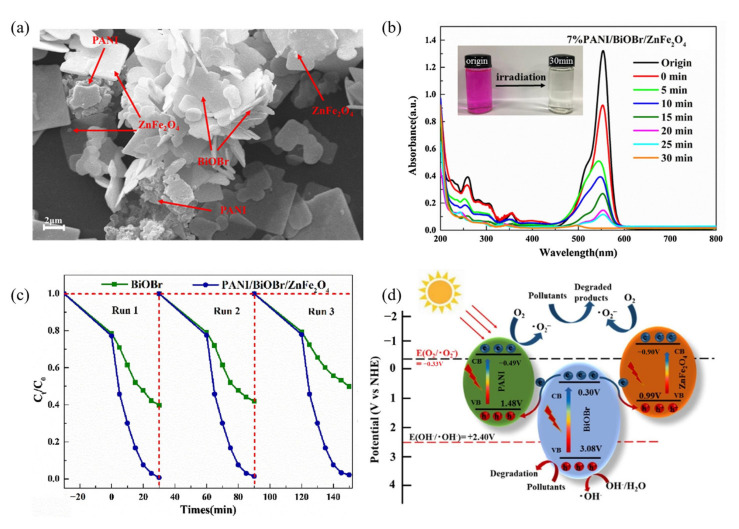
PANI/BiOBr/ZnFe_2_O_4_: (**a**) scanning electron microscopy (SEM) image; (**b**) the UV-Vis absorption spectra changes of RhB solution during the photocatalytic process over 7% PANI; (**c**) the cyclic photodegradation curves; (**d**) dual Z-scheme electron transfer and pollutant degradation mechanism under visible-light exposure [102] (reprinted with permission from [102] Zhang et al., 2020).

**Figure 12 ijerph-19-10710-f012:**
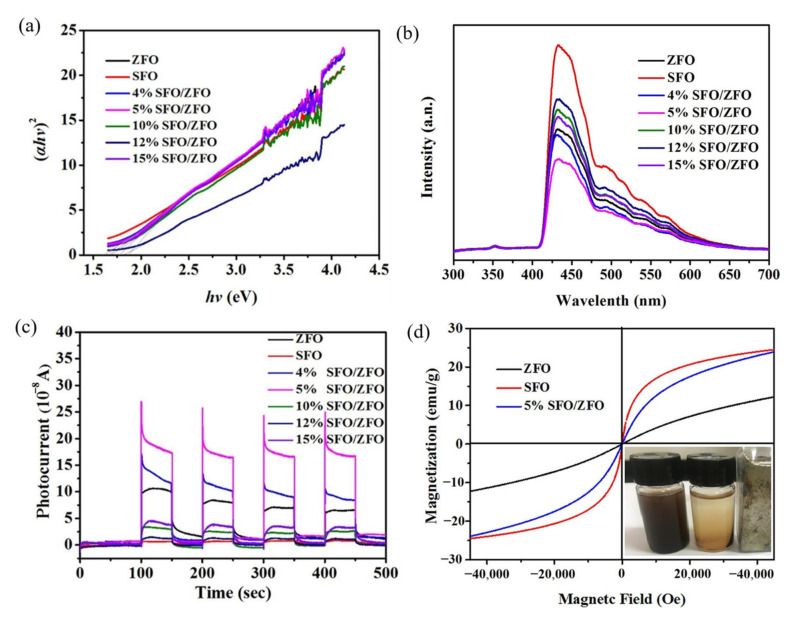
The pure ZnFe_2_O_4_, pure SnFe_2_O_4_, and SnFe_2_O_4_/ZnFe_2_O_4_ nano-heterojunctions: (**a**) plots of (ahv)^2^ versus hv; (**b**) PL spectra; (**c**) photocurrent responses; (**d**) magnetization curve [103] (reprinted with permission from [103] Wang et al., 2020).

**Figure 13 ijerph-19-10710-f013:**
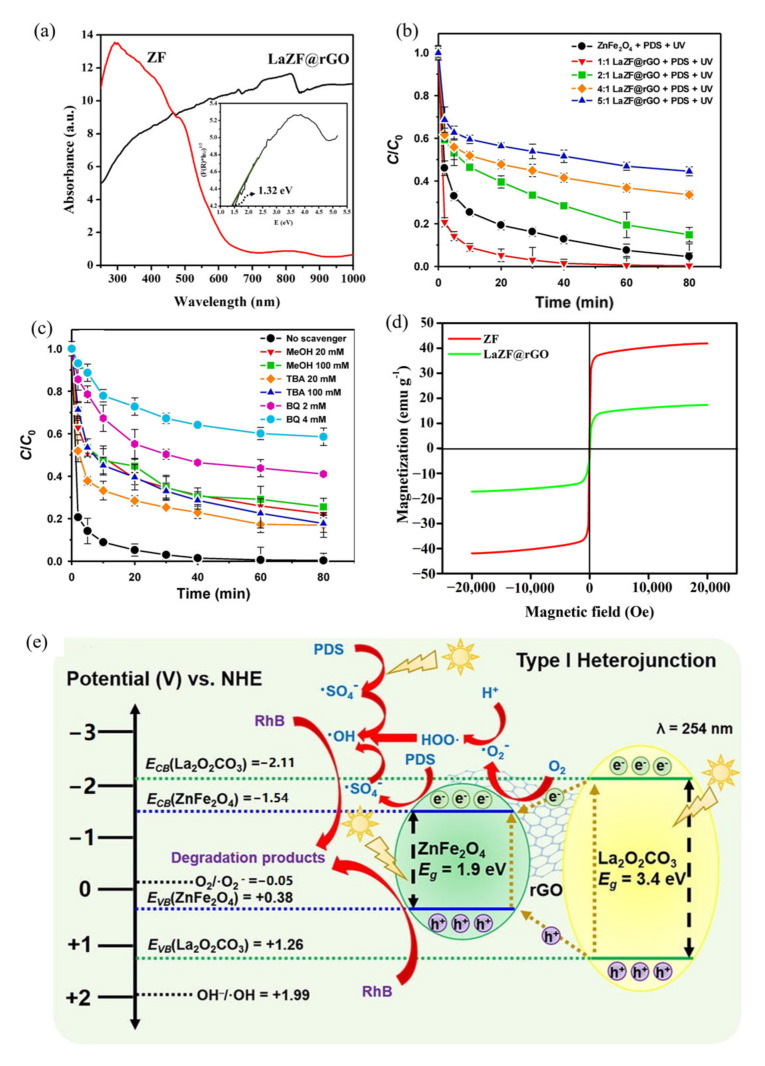
ZnFe_2_O_4_ and LaZF@rGO: (**a**) UV-Vis absorption spectra; (**b**) influence of La content in LaZF@rGO; (**c**) radical scavengers on RhB decolorization; (**d**) Magnetization curve; (**e**) a schematic representation of the RhB degradation mechanism [108] (reprinted with permission from [108] Jun et al., 2020).

**Figure 14 ijerph-19-10710-f014:**
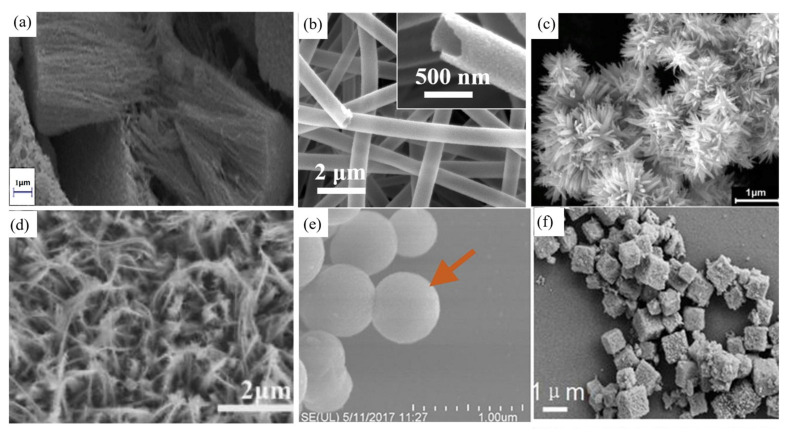
The SEM images of ZnFe_2_O_4_ with different morphologies: (**a**) nanowire (field emission SEM) [112] (reprinted with permission from [112] Ghazkoob et al., 2022); (**b**) hollow nanofibers [113] (reprinted with permission from [113] Liu et al., 2019); (**c**) flower-shaped nanorods [91] (reprinted with permission from [91] Wang et al., 2019); (**d**) nanowire arrays [114] (reprinted with permission from [114] Chen et al., 2020); (**e**) yolk-shell [99] (reprinted with permission from [99] Li et al., 2019); (**f**) hollow cube skeleton [115] (reprinted with permission from [115] Cao et al., 2018).

**Figure 15 ijerph-19-10710-f015:**
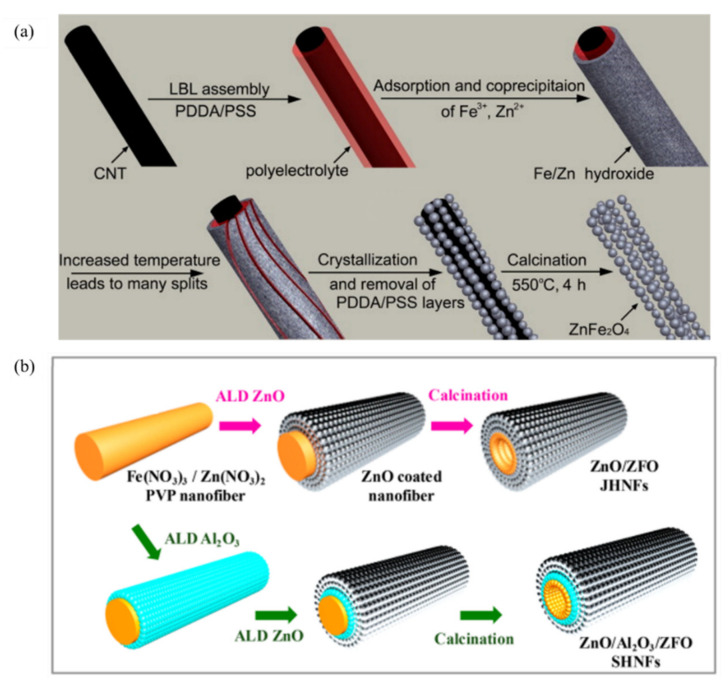
(**a**) A schematic representation of the synthesis process of ZnFe_2_O_4_ nanochains [116] (reprinted with permission from [116] Zhang et al., 2010). (**b**) A schematic representation of the synthesis process of the ZnFe_2_O_4_ nanochains [113] (reprinted with permission from [113] Liu et al., 2019).

**Figure 16 ijerph-19-10710-f016:**
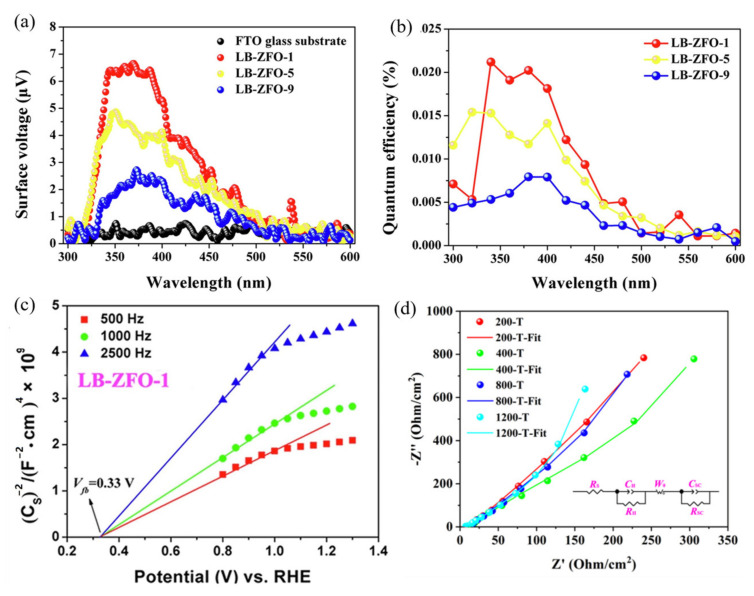
The LB-ZFO-1, LB-ZFO-5 and LB-ZFO-9 films: (**a**) surface photovoltage; (**b**) apparent quantum efficiency; (**c**) Mott–Schottky plots of LB-ZFO-1; (**d**) electrochemical impedance spectroscopy spectra [117] (reprinted with permission from [117] Zhu et al., 2020).

**Figure 17 ijerph-19-10710-f017:**
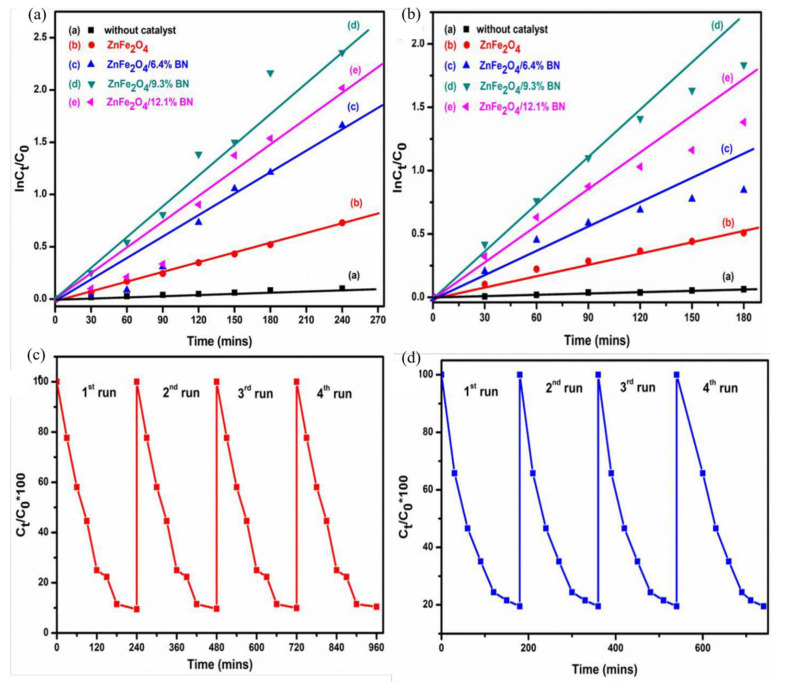
The kinetic plots in the photodegradation of (**a**) CR and (**b**) TC by the as-synthesized catalysts; photo-stability of the ZnFe_2_O_4_/9.3% BN composite catalyst in the degradation of (**c**) CR and (**d**) TC [120] (reprinted with permission from [120] Erusappan et al., 2020).

**Figure 18 ijerph-19-10710-f018:**
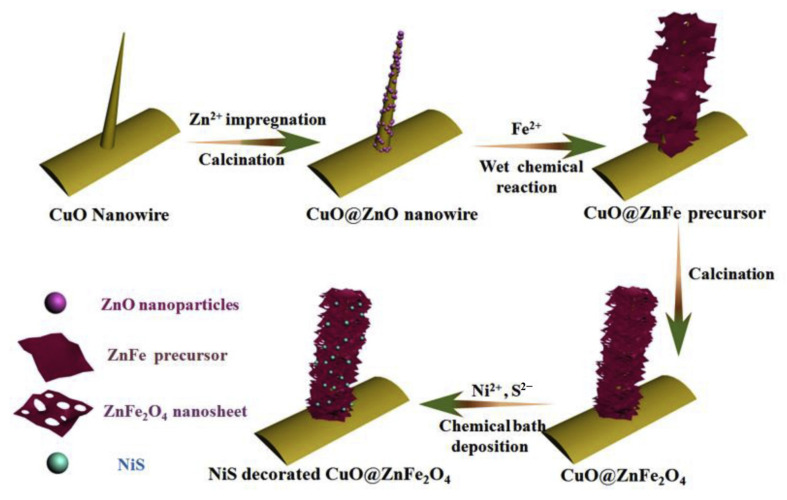
A schematic representation of the formation process of the CuO@ZnFe_2_O_4_ core/shell nanoarrays decorated with NiS quantum dots [114] (reprinted with permission from [114] Chen et al., 2020).

**Figure 19 ijerph-19-10710-f019:**
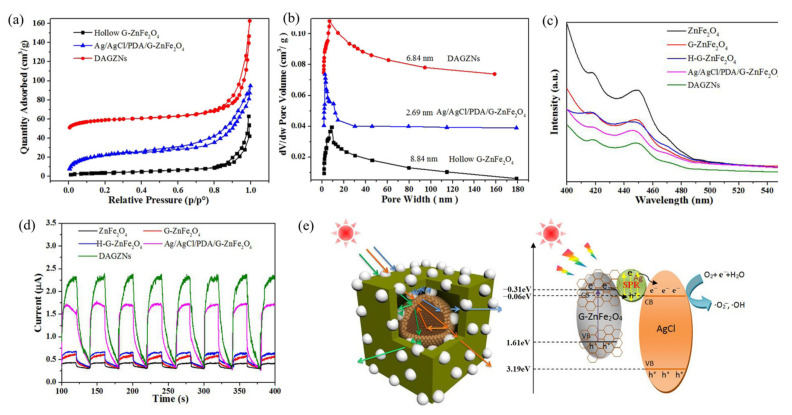
(**a**) The nitrogen adsorption–desorption isotherms of hollow G-ZnFe_2_O_4_, Ag/AgCl/PDA/G-ZnFe_2_O_4_, and DAGZNs; (**b**) Barrett–Joyner–Halenda (BJH) pore size distribution; (**c**) PL spectroscopy; (**d**) transient photocurrent; (**e**) schematic representation of the light scattering effect in DAGZN [125] (reprinted with permission from [125] Zhang et al., 2020).

**Table 1 ijerph-19-10710-t001:** A comparison of the photodegradation rate of organic pollutants by ZnFe_2_O_4_ with different preparation methods.

ZnFe_2_O_4_ Dose (g/L)	Synthesis Method	Pollutants and Concentrations (mg/L)	Time (min) and Degradation Efficiency (%)	Refs.
0.2	Biosynthesis method (Nyctanthes arbor-tristis flower extract)	RhB, 5	30, 100	2019 [68]
0.5	Co-precipitation	Methylene Blue (MB), 10	50, 99	2018 [72]
——	Biosynthesis method (Jatropha extract)	Sudan, ——	——, 98	2020 [69]
——	Biosynthesis method (Simbang Darah (Iresine herbstii) extract)	Direct Red 81, 35	120, 99.66	2021 [70]
0.5	Biosynthesis method (Sugarcane juice)	MB, 5 & RhB, 5	120, 98 & 95	2018 [73]
0.6	——	Terramycin, 40	60, 94.68	2022 [74]
2	A single-mode microwave combustion process	RhB, 10	60, 5	2022 [75]
0.3	Electrospinning and co-precipitation	MB, 5	60, 29.7%	2022 [76]
1	Hydrothermal	MB, 2	240, 20	2018 [77]
0.1	Ultra-sonication assisted chemical precipitation method	MB, 25	160, 66.4%	2021 [78]

**Table 2 ijerph-19-10710-t002:** The principles and the advantages and disadvantages of the different methods used to prepare the ZnFe_2_O_4_ photocatalysts.

Method	Preparation Principles	Advantage	Shortcoming
Microwave-assisted method	Using microwave heating, energy is directly transferred to the reactant particles through chemical interaction with electromagnetic radiation, providing a thermally uniform nucleation environment.	The product has a very fine particle size, a high heating rate, high purity, low thermal inertia, and low cost.	Complex equipment, different reactors are needed to apply high pressure, and the reactors take a long time to react.
Ball milling method	Dry or wet milling of the metal salt mixture in a ball mill breaks up and combines the particles repeatedly.	Low cost, simple operation, no use of toxic and expensive solutions, prevention of particle agglomeration, potential for large-scale production.	Low product purity, risk of material contamination during milling, long production cycle, special equipment required.
Vapor deposition method	Using thermal decomposition, precursor gases are converted to thin films, which are deposited on substrates.	It has good conformal coverage on complex structures and can deposit various materials such as metals, oxides, semiconductors, etc.	Crystallinity of the deposited layer is usually columnar and not bendable.
Sonochemical method	Mechanical waves act on the fluid, producing a sharp motion that results in tiny bubbles that compress the solvent and sparse it, thereby speeding up chemical reactions.	Low energy consumption; quick response; uniform size; simple and economical.	The product has a low magnetic moment and poor crystallinity.
Pulsed laser method	Different substrates are bombarded with laser light, and the bombarded substances are deposited.	High deposition rate, short test cycle, easy cleaning.	Molten particles generated by the laser acting on the target can cause contamination of the material and uneven film thickness.
Ceramic method	After the two metal oxides (ZnO and Fe_2_O_3_) are mixed and ground into fine particles, they are pressed into particles using a press and sintered at a high temperature.	——	Some of the reactants and intermediate products remain. The purity is not high, the particle size is large, the reaction temperature is extremely high, the limitations are obvious, and the energy consumption is large.
Spray pyrolysis	The raw materials are first dissolved and then atomized and volatilized to obtain the product.	As raw materials are thermally decomposed during the preparation process, the reaction temperature is low, operation is quick and simple, and washing and grinding are unnecessary.	Equipment is easily corroded by polluted reaction gases.

**Table 3 ijerph-19-10710-t003:** The band gap change in the ZnFe_2_O_4_ photocatalysts after doping at different sites.

Researcher	A Site	B Site	Modified Catalyst	E_g_/eV	Refs.
Son et al.	Ni	--	ZnFe_2_O_4_	1.81	[89]
			Zn_0.75_Ni_0.25_Fe_2_O_4_	1.73	
			Zn_0.4_Ni_0.6_Fe_2_O_4_	1.72	
Somvanshi et al.	Mg	--	ZnFe_2_O_4_	2.39	[62]
			Zn_0.5_Mg_0.5_Fe_2_O_4_	1.96	
Fadl et al.	Cr, Ni	--	Ni_0.4_Zn_0.5_Cr_0.1_Fe_2_O_4_	3.9	[90]
			Ni_0.4_Zn_0.3_Cr_0.3_Fe_2_O_4_	3.85	
			Ni_0.4_Cr_0.6_Fe_2_O_4_	3.78	
Abdo et al.	Co, Cu	Y	Co_0.5_Cu_0.25_Zn_0.25_Fe_2.0_O_4_	1.57	[91]
			Co_0.5_Cu_0.25_Zn_0.25_Y_0.06_Fe_1.94_O_4_	1.54	
			Co_0.5_Cu_0.25_Zn_0.25_Y_0.1_Fe_1.9_O_4_	1.52	
Lemziouka et al.	--	La	ZnFe_1·99_La_0·01_O_4_	1.945	[92]
Tholkappiyan et al.	--	La	ZnFe_1·94_La_0·06_O_4_	1.97	[93]
Shoba et al.	--	Er	ZnFe_2_O_4_	1.85	[94]
			ZnFe_1.8_Er_0.2_O_4_	1.93	
			ZnFe_1.6_Er_0.4_O_4_	1.96	

**Table 4 ijerph-19-10710-t004:** The degradation performance of the ZnFe_2_O_4_ photocatalyst contaminants using different modification methods.

Sr. No.	Catalyst and Dose (g/L)	Modification Method	Light Source	Pollutants and Concentrations (mg/L)	Time (min) and Degradation Efficiency (%)	Stability	Refs.
1	Co_0.5_Zn_0.5_Fe_2_O_4_, 0.05	Elemental doping (A site)	300 W Tungsten lamp	MB, 10	60 min, 77%	——	[127]
2	Ni_0.1_Zn_0.9_Fe_2_O_4_, 0.4	Elemental doping (A site)	500 W Xenon lamp	Levofloxacin, 10	90 min, 98.6%	Five cycles 100%	[133]
3	Zn_0.9_Dy_0.1_Fe_2_O_4_, 1	Elemental doping (A site)	125 W Mercury lamp	MB, 20	45 min, 97.3%	Five cycles 100%	[134]
4	Co_0.5_Cu_0.25_Zn_0.25_Y_0.1_Fe_1.9_O_4_, 0.5	Elemental doping (AB site)	250 W Mercury lamp and 500 W Xenon lamp	MB, 20	60 min, 95%	Five cycles 99%	[91]
5	Zn_0.5_Cu_0.5_Fe_1.95_Ce_0.05_O_4_,1	Elemental doping (AB site)	250 W Mercury lamp and 500 W Tungsten lamp	Bisphenol A, 10	180 min, 91.3%	Three cycles 97%	[55]
6	Cd_0.75_Zn_0.25_Co_0.25_Fe_1.75_O_4_, 0.2	Elemental doping (AB site)	400 W Mercury lamp	MO, 2	120 min, 82%	Six cycles 100%	[128]
7	ZFCN@20PPY, 0.2	Compound modification (conducting polymers)	Solar light	Ciprofloxacin, 20	120 min, 92%	Four cycles 100%	[101]
8	PANI/BiOBr/ZnFe_2_O_4_, 1	Compound modification (conducting polymers)	Mercury lamp	RhB, 20	60 min, 99%	Three cycles 98%	[102]
9	ZnFe_2_O_4_/SnFe_2_O_4_, 0.3	Compound modification (metal oxide)	300 W Xenon lamp	Pharmaceutical wastewater, ——	720 min, 77.5%	Four cycles 100%	[103]
10	ZnFe_2_O_4_/TiO_2_, 0.83	Compound modification (metal oxide)	150 W Mercury lamp	RhB, 50	10 min, 97.87%	——	[135]
11	ZnFe_2_O_4_/Fe_2_O_3_, 0.5	Compound modification (metal oxide)	300 W Xenon lamp	Ciprofloxacin, 10	180 min, 96.5%	Five cycles 100%	[56]
12	CeO_2_/ZnFe_2_O_4_, 0.15	Compound modification (metal oxide)	500 W Tungsten lamp	MB, 25	90 min, 90.48%	Six cycles 92%	[104]
13	γ-MnO_2_@ZnFe_2_O_4_/rGO, 0.2	Compound modification (metal oxide and carbon material)	——	Phenol, 20	30 min, 100%	Five cycles 100%	[131]
14	ZnFe_2_O_4_/SnS_2_, 0.03	Compound modification (metal sulfide)	500 W Tungsten lamp	MB, 25	45 min, 91.78%	Six cycles 100%	[105]
15	ZnFe_2_O_4_/CdS, 1	Compound modification (metal sulfide)	Visible light	RhB, 5	120 min, 100%	Three cycles 100%	[106]
16	LaZF@rGO, 0.25	Compound modification (carbon material)	11 W Tungsten lamp	RhB, 30	60 min, 100%	Four cycles 91%	[108]
17	ZnFe_2_O_4_@rGO, 0.5	Compound modification (carbon material)	150 W One-sun solar simulator	MB, 10	60 min, 94%	Three cycles 91%	[109]
18	C-ZnFe_2_O_4_, 1.5	Compound modification (carbon material)	——	Noroxin, 5	40 min, 86.2%	Four cycles 98%	[111]
19	ZnFe_2_O_4_, ——	Morphology modification (nanosheets)	——	Sudan red, ——	120 min, 67%	——	[69]
20	ZnFe_2_O_4_, 0.25	Morphology modification (nano stave)	150 W UV Osram lamp	MB, 5	45 min, 99.8%	Five cycles 100%	[136]
21	ZnFe_2_O_4_/BiVO_4_, 0.6	Morphology modification (nanowires)	100 W Xenon lamp	CR, 20	105 min, 100%	——	[112]
22	ZnO/Al_2_O_3_/ZnFe_2_O_4_, 1	Morphology modification (nanofibers)	150 W Xenon lamp	MB, 10	120 min, 90%	——	[113]
23	ZFO/FTO, ——	Morphology modification (films)	350 W Xenon lamp	MB, 100	19 min, 100%	Ten cycles 100%	[119]
24	ZnFe_2_O_4_, 0.5	Morphology modification (hollow cube)	300 W Xenon lamp	Tetracycline hydrochloride, 40	20 min, 80%	——	[115]

## Data Availability

Not applicable.

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
