# Peer review of "Progress in the Preparation and Modification of Zinc Ferrites Used for the Photocatalytic Degradation of Organic Pollutants"

_ijerph, 2022, doi:10.3390/ijerph191710710_

Round 1

Reviewer 1 Report

This paper describes a Progress in the Preparation and Modification of Zinc Ferrites  Used for the Photocatalytic Degradation of Organic Pollutants. It seems interesting. I think it can be consider to publish if the following issues are solved:

1. some recent achievements should be summarize in one figure including the recent applications and the modification methods like the morphology or defect engineering.

2. The preparation methods also should be discussed in detail 

3, some recent revalent papers should be cited: Small Methods 2200314 (2022) doi:10.1002/smtd.202200314; Colloids and Surfaces A: Physicochemical and Engineering Aspects 646, 128962.

4. the introduction should add some recent achievement of this kind of materials.

5. the english level should be improved.

Reviewer 2 Report

          This review focused on the preparation and modification of Zinc Ferrites used for the photocatalytic degradation of organic pollutants. Photocatalytic remediation of zinc ferrite is a type of photocatalytic material with high physicochemical stability, narrow band gap, high carrier separation ability, high porosity, and paramagnetism, which makes it easy to recover.   The topic of the review article is important and the structure of the paper is nice but some issues should be addressed before considering the publication of the paper : (minor revision)

 1.    The authors should explain the importance of various organic pollutants and their remediation methods in the introduction part of the manuscript.

 2.    The author has focused on the biosynthesis method, which can add compare the green synthesized nanoparticle with photocatalytic remediation. This could give a clear idea to the reader.

 3.    The photocatalytic reusability test and different light intensity investigations can include in the review manuscript for a clear understanding of the readers.

 4.    There are some scientific, grammatical and typo errors in the text that need to be re-checked and corrected more carefully.

 5.    From this review work on organic pollutants remediation, it is understood that ZnFe2O4 shows better catalytic activity. The suggestion to the author, carry out the effect of catalytic dose and the effect of dye concentration of the photocatalytic reaction. This will be clearer to readers.

 6.    Recent literature reported in the New Journal of Chemistry (10.1039/D0NJ01272F) and Optik (10.1016/j.ijleo.2021.166599) should be cited in the manuscript. 

Reviewer 3 Report

Review comments to the author

Title: Progress in the Preparation and Modification of Zinc Ferrites Used for the Photocatalytic Degradation of Organic Pollutants

Manuscript ID: ijerph-1854177

This review is related to the use of zinc ferrites in photocatalytic degradation of organic pollutants. Various preparation methods of zinc ferrite is also described.

I have noticed some inaccuracies in the text. I ask the authors to carefully verify English style.

In my opinion, the manuscript needs major revision. Below I have outlined some observations and suggestions:

- page. 2, line 73 – „... Nair temperature ..”? Maybe, Neel ...

line 55 – The complex .... instead of .. The Complex (not C capital)

line 68 - In recent years instead of While in recent years

line 75 - 250 °C  instead of 250°C

line 77 - 1.9 eV instead of 1.9eV

- page 3, line 91 – I suggest ... groups of research ... instead of ... body of research ...

Fig. 2 –  improvement of the resolution is suggested.

line 106 – The photocatalytic ... instead of .. A photocatalytic ...

- page 4, line 122 – I suggest ... saturation magnetization value (or magnitude) ... instead of .. saturation magnetization intensity (it is not the most appropriate expression). The same observation in lines 138 and 145, 199, 202, 266, 280, 290

line 131 – I suggest ... the solvent strongly affects... instead of .. the solvent strongly effects

line 141 – By the hydrothermal process does not use less Zn ions than by the ceramic method. As it is described in reference 40, the Fe/Zn atomic ratio is 2:1. This phrase should be reformulated.

- page 5, line 158 –5 h instead of 5h

line 159, 160 – I suggest ... exhibiting superparamagnetism behavior ... instead of .. exhibiting superparamagnetism

- page 6, line 174 – what means „concentration of ZnFe2O4”. Maybe, concentration of metal ions? Or Zn2+ concentration?

lines 175, 176 – it should be reformulated to avoid repetition „.. due to the production of α-Fe2O3 due to insufficient ..”

line 178 – it should be reformulated to avoid repetition „..upon increasing the pH to 13 upon adding ..”

line 184 – I suggest the reformulation: „This method consists in the formation of gels by ...” ... instead of … This method forms gels …

- page 9, point (3), lines 245, 246 – it should to be reformulated

- page 11, line 296 - ... structure could produce ... instead of ... structure could also produce ...

line 303 - ... Figure 8 shows that the band gap .. instead of ... Figure 8 shows the band gap ...

- page 12, line 360 – the (not capitals) VB

Fig. 9e – improvement of the resolution is suggested.

- page 13 - Fig. 10d – the letters would be more visible

line 379 – „... and migration in fish materials ...”- what means fish materials? Composite? It should be replaced with another expression.

- page 14, line 386 – „Shaman et al. [77] prepared CeO2/ZnFe2O4 using this method. The ...”- what method exactly? Please, reformulate!

- page 15 - Fig. 12b – The meaning of OX axis (Time?) is missing

- page 19, lines 559, 560 – „The PL intensity shown in Figure 17(c) indicates the PL intensity of...” –  The PL intensity indicates the PL intensity? It makes no sense. Please, reformulate!

- page 20 – „Figure 17. SEM images of ZnFe2O4 prepared with different morphologies ...„. (see caption)? Figure 17 does not contain SEM images.

References section – the title of the journal must be abbreviated (see the Instructions for authors). Please, check all references!

Reference 39: the title of the article – not capitals

Best regards

Round 2

Reviewer 1 Report

It is well revised and can be accepted now.

Reviewer 3 Report

Dear Authors,

I have read the revised form of you article and I agree with its publication in IJERHP.